# From Hype to Reality: Unveiling the Promises, Challenges and Opportunities of Blockchain in Supply Chain Systems

**Muen Uddin [1,*], Shitharth Selvarajan [2], Muath Obaidat [3], Shams Ul Arfeen [4], Alaa O. Khadidos [5], Adil O. Khadidos [6] and Maha Abdelhaq [7]**

1. College of Computing and Information Technology, University of Doha for Science and Technology, Doha 24449, Qatar
2. Department of Computer Science, Kebri Dehar University, Kebri Dehar 250, Ethiopia; shitharths@kdu.edu.et
3. Department of Computer Science, City University New York, New York, NY 10036, USA; muobaidat@ccny.cuny.edu
4. National Advanced IPv6 Centre (NAv6), Universiti Sains Malaysia, Gelugor 11800, Malaysia; shamsularfeen@nav6.usm.my
5. Department of Information Systems, Faculty of Computing and Information Technology, King Abdulaziz University, Jeddah 21589, Saudi Arabia; aokhadidos@kau.edu.sa
6. Department of Information Technology, Faculty of Computing and Information Technology, King Abdulaziz University, Jeddah 21589, Saudi Arabia; akhadidos@kau.edu.sa
7. Department of Information Technology, College of Computer and Information Sciences, Princess Nourah bint Abdulrahman University, P.O. Box 84428, Riyadh 11671, Saudi Arabia; msabdelhaq@pnu.edu.sa
* Correspondence: mueen.uddin@udst.edu.qa

**Abstract:** Blockchain is a groundbreaking technology widely adopted in industrial applications for improving supply chain management (SCM). The SCM and logistics communities have paid close attention to the development of blockchain technology. The primary purpose of employing a blockchain for SCM is to lower production costs while enhancing the system's security. In recent years, blockchain-related SCM research has drawn much interest, and it is fair to state that this technology is now the most promising option for delivering reliable services/goods in supply chain networks. This study uses rigorous methods to review the technical implementation aspects of SCM systems driven by Blockchain. To ensure the security of industrial applications, we primarily concentrated on developing SCM solutions with blockchain capabilities. In this study, the unique qualities of blockchain technology have been exploited to analyze the main effects of leveraging it in the SCM. Several security metrics are utilized to validate and compare the blockchain methodologies' effectiveness in SCM. The blockchain may alter the supply chain to make it more transparent and efficient by creating a useful tool for strategic planning and enhancing connections among the customers, suppliers, and accelerators. Moreover, the performance of traditional and blockchain-enabled SCM systems is compared in this study based on the parameters of efficiency, execution time, security level, and latency.

**Keywords:** supply chain management (SCM); blockchain; privacy preservation; logistics; hyperledger; smart contract

## 1. Introduction

The manufacturing industry now demands a sustainable environment and an effective balance of societal and financial goals [1,2]. Manufacturing companies can now meet increasing customer demands while having little negative impact on the ecosystem and community. With the significance of sustainable practices, manufacturers are adopting various technologies and methods to improve their sustainability performance in the market, including big data analytics, artificial intelligence, and blockchain [3,4]. Blockchain is one of the newest and fastest-growing digital technologies that can support sustainable manufacturing. Blockchain technology uses a distributed and independent data structure,

which enables data sharing on a public network. In Blockchain [5–7], every transaction is authorized, confirmed, and made accessible by other network users [5–7].

Moreover, the Blockchain guarantees the security and integrity of the transactions [8]. Blockchain was first used in the banking sector to replace manual validation of transactions with digital credentials for cryptocurrencies [9,10]. Blockchain has drawn interest from academics and industry experts because it might improve accountability, trust, openness, data consistency, and security. In general, blockchain implementations can be either private or public. In addition, it has the potential to provide complete transparency throughout international supply chains. Consequently, it can result in better tracking of items and offer tamper-proof data to create trust between stakeholders. The supply chain [11] industries can adopt more sustainable practices due to the potential benefits of blockchain technology.

### 1.1. Types of Blockchain

Blockchain is generally categorized into three types: public, private, and consortium, in which public Blockchain is a permissionless network accessible to everyone. Anyone can join the network and engage in this kind of Blockchain by learning, generating content, or interacting outside it. A public blockchain has decentralized features and lacks a central authority that governs the entire network. Moreover, the data on a public blockchain are safe since it is impossible to change or edit data once verified on the Blockchain. A private blockchain is permissioned since a network operator manages it, and only approved users can join the network. The network is controlled by a few entities, which makes it necessary to conduct transactions through external parties. Only the parties involved in the transaction will know about it in this type of Blockchain; others will not be able to access it, making the transaction private. A hybrid blockchain combines elements from both private and public Blockchain. It seeks to balance the two blockchain techniques' advantages while minimizing their drawbacks. It has applications for businesses deploying private and public Blockchains to benefit from the integrated benefits. A private permission-based system and a public permission-less system are both possible with such a Blockchain network. The other type is a consortium blockchain, which aims to eliminate the private Blockchain's single-entity independence.

Contrary to a private blockchain, more than one entity exists on the network in the event of a consortium blockchain. The decentralized nature of the control is maintained because no single authority controls it. The core features of these blockchain techniques are illustrated in Figure 1a, and the difference between the types of blockchain technologies are illustrated in Figure 1b. The major drawbacks of using private blockchain are given in below:

- Establishing trust: A private network has fewer users than a private network.
- Reduced security: Since there are a smaller number of nodes or participants in a private blockchain network, hence it is more susceptible to a security breach.
- The limitation of private block chains is their need on a central Identity and Access Management (IAM) solution. This system offers complete managerial and tracking capabilities.

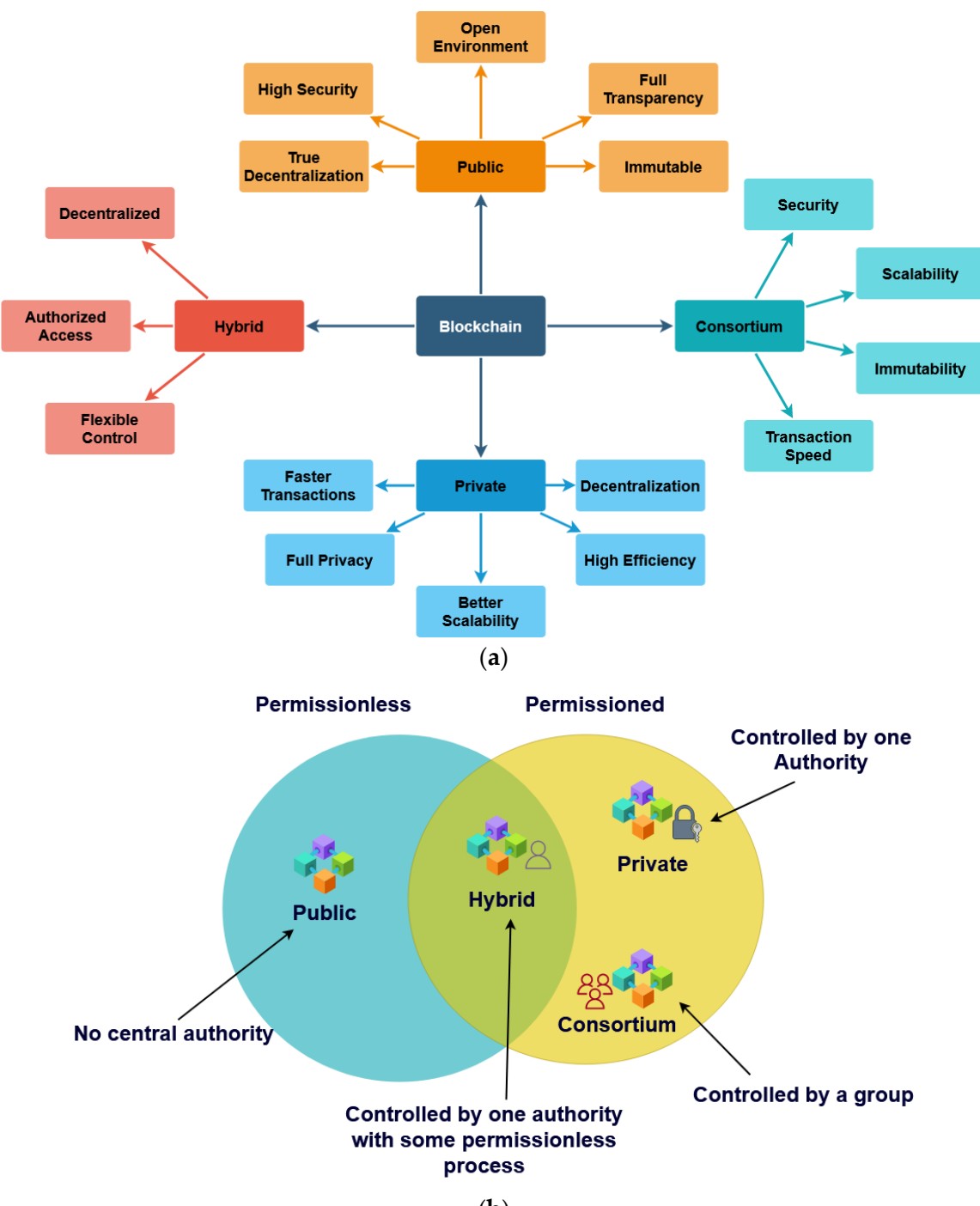

**Figure 1.** (**a**). Features of Blockchain. (**b**). Difference between various blockchain technologies.

### 1.2. Supply Chain Management (SCM)

The supply chain management (SCM) and logistics communities have paid close attention to the development of blockchain technology [12]. Supply chain management typically involves managing and preserving producer relationships with vendors, administration, and customers to provide higher customer value at a lower cost. The notion of the Internet of Things (IoT), which enables the use of numerous intelligent actuators and sensors connected to the Internet, provides creative ways for monitoring the movement of goods and commodities essential to supply chains [13,14]. It specifically addresses the essential element of supply chain management—the seamless exchange of information

about a product between parties involved in its entire life cycle. Figure 2 shows the uses of Blockchain in SCM.

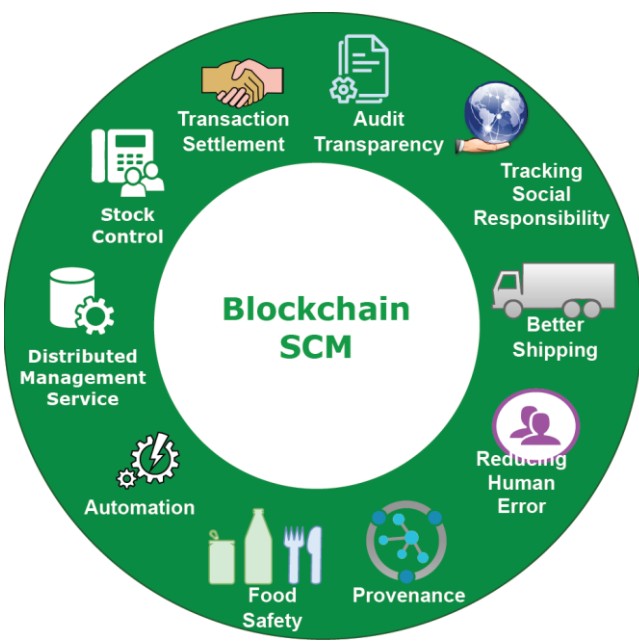

**Figure 2.** Blockchain SCM.

For the management of distribution centers, reviewing the past transactions will help them have confidence in the supply reliability and optimize the supply chain; therefore, the issue of ensuring the reliability and openness of the supply chain is equally crucial for intermediate suppliers. Due to rising competition in the market for logistics mediators and the need for new, creative ways of working with digital technologies, the concerns of supply chain management optimization and improvement are also pertinent and in demand. The capacity of digital technologies to radically alter the current organizational and financial procedures of supply chain management. The linear supply chains are already evolving into dynamic, interconnected open supply systems (digital supply networks), where information flows are perpetual and readily available to all network customers, allowing the avoidance of various operational issues and delays present in the traditional supply chains. One of these modern digital technologies is the blockchain. Due to the rapid expansion, it has just lately become more widely used. In the beginning, it was employed to hide transaction information. Presently the blockchain technology is rapidly rising in popularity. On its foundation, startups are being presented in many different economic sectors. It is the subject of media outlets, forums, and industry-specific conferences.

The main objectives of this study are as follows:

- The technical implementation facets of SCM systems powered by Blockchain are reviewed in this study using a methodical approach.
- We mainly focused on developing blockchain-enabled SCM systems to ensure the security of industrial applications.
- The significant impacts of using blockchain technology in the SCM have been analyzed in this study with its distinct properties.
- The performance of the recent blockchain methodologies used in the existing works is validated and compared using several security parameters.

The following subsections comprise the remaining parts of this article: The comprehensive literature evaluation of blockchain-enabled SCM systems is presented in Section 2. The digital ledger technology, use in SCM, and blockchain-based privacy preservation methods are all clearly explained in Section 3. The performance findings and comparison analysis among the existing blockchain-enabled SCM approaches are validated in Section 4.

Moreover, the current challenges and future recommendations for the Blockchain SCM are stated in Section 5. Finally, Section 6 summarizes this study, including its recommendations, outcomes, and future actions.

## 2. Knowledge and Related Works

Recent research has covered various blockchain applications in SCM and logistics [15,16]. Several aspects of implementing Blockchain in logistics and SCM have been investigated, starting with conceptualizing its potential promises and usage scenarios. It could be more helpful for organizations to explore the effects of connectivity barriers on supply chain interaction and adaptation. Blockchains [17] can revolutionize supply chain planning, structure, management, and operations. A significant rethinking of supply chain management is predicted by the Blockchain's capacity to guarantee the dependability, accountability, and legitimacy of information and smart contract associations for an untrustworthy setting [18].

Numerous logistics operations are included in supply chain processes, including a successful organization, execution, and management of the flow and storage of goods/services and associated information from the point of origin to the point of usage to meet customer demands [19]. Blockchain-enabled SCM takes advantage of accessibility, revenue optimization, inventory turnover, supply chain speed, and efficient customer service delivery by combining and automating these activities. Supply chains [20] are complicated and exposed to various uncertainties and dangers, including the involvement of suppliers in opportunistic behavior (such as falsifying information or stealing), security breaches, online criminal activity, and the detection of counterfeit goods. Corporate executives from various industries desire to enhance SCM through digitization to get around these problems. Organizations can collaborate and conduct business with their trading counterparts and chains by implementing inter-organizational systems [21], called the digitization of supply chains. It is essential to consider potential applications that could persuade supply chain managers to adopt this technology as an authorized provider for sharing knowledge and data because supply chain management research on Blockchain remains in its early stages [22]. Before implementing a blockchain-enabled supply chain in a real-world setting, a proof of concept must be demonstrated empirically for business enhancement. It is still possible for data that businesses upload to the Blockchain to be faked or wrong in the future; therefore, the Blockchain's legitimacy is still heavily reliant on the trustworthiness of its partner businesses. Verification is required to demonstrate that participants in a supply chain [23,24] cannot function at a high level if they communicate demand and inventory data that is false or erroneous on the Blockchain.

More factors affect product and material flows in the supply chain. Each item may have a presence on the Blockchain in the digital world, allowing all necessary parties to have immediate access to the product profile. Security measures can restrict access so only those with the proper digital keys can use a specific product. Many other types of data can be gathered, including information about the product's status, the environment, and the standards that need to be followed [25]. An identifier that connects tangible goods to their digital identity in the Blockchain is represented by a data tag linked to a product [26]. The way a product is 'owned' or exchanged by a particular actor is a fascinating structure and process control feature. A critical regulation requires consent from actors before they can start a trade or add new information to a product's profile. This consent might be obtained through smart contract deals and agreements [27]. Both parties can sign a digital agreement or satisfy a smart contract requirement to validate the transaction before a product is delivered to another actor. The blockchain ledger is updated with transaction information after all participants have complied with their duties and procedures [28]. The system will automatically update the data transaction records when a change is made. A minimum of five key elements of a product can be emphasized and explained with blockchain technology: its character, grade, quantity, position, and property.

Customers can view the continuous chain of provenance and transactions from the initial supply to the sale of the goods through the Blockchain [29], eliminating the demand for

a reliable centralized body to manage and maintain this system. Ledgers are used to record transactions' effect on each of those blockchain data parameters, and they are updated with verifiable information. With computerized regulations, blockchain dependability [30] and transparency are designed to efficiently enable the flow of materials and information along the supply chain. Figures 3 and 4 show the traditional and blockchain-enabled SCM framework models. Compared to the conventional SCM framework, the blockchain-enabled SCM provides enormous properties to the users. Smart contracts, which define the rules maintained on the Blockchain, can aid in determining how network actors interact with one another and the system as a whole. Smart contracts among supply chain actors influence network data exchange and continual process improvement. On the network, performers and goods have a digital profile with details such as an overview, location, identification, and affiliation with particular goods.

Using blockchain technology, Cole et al. [31] implemented an operations and supply chain management (OSCM) strategy. This work aims to ensure the safety and security requirements with assured quality. Bencic et al. [32] utilized a distributed ledger technology incorporated with the smart tag for protecting IoT supply chain systems. The TagItSmart (TIS) system is a connected device logistics management solution that provides products with Smart Tags and allows information sharing between stakeholders who participate in a product lifetime using smart labels. Typically, the distributed ledger technology makes it possible for a group of individuals operating in a distributed setting who are mutually untrustworthy to maintain an overseas, append-only data structure. Distributed ledgers stand out for their stability, resistance to control, distributed service, and the absence of a centrally located authorized third party.

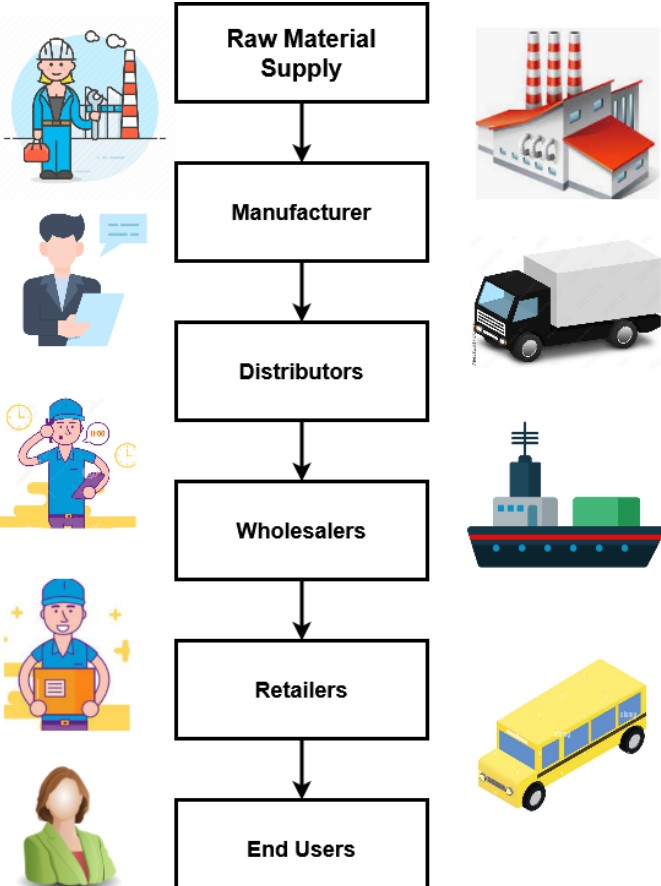

**Figure 3.** Traditional supply chain system.

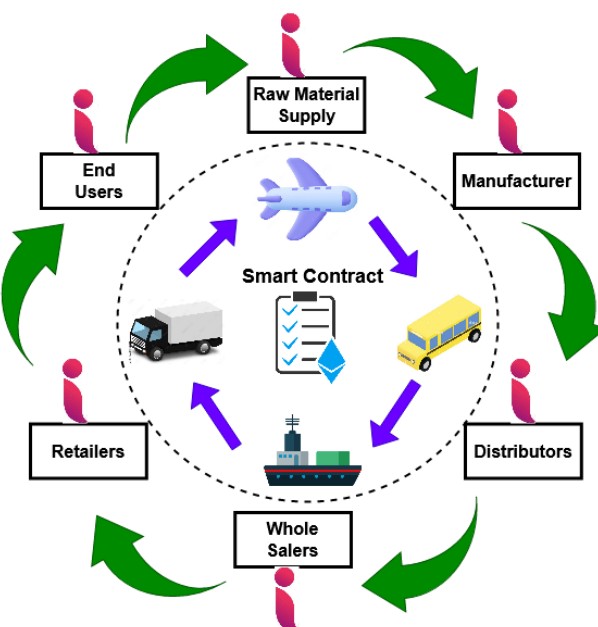

**Figure 4.** Blockchain-based supply chain system.

Jabbar et al. [33] presented a comprehensive analysis to examine the challenges and future work in blockchain-enabled supply chain systems. To convert raw materials and natural assets into completed goods that can be delivered to the final consumer, the supply chain is an interconnected system of businesses, tasks, materials, individuals, and information. Hyper-segmentation, localized production of products, suppliers, and consumers are some of the leading forces transforming the conventional supply chain. Traceability can guarantee a product's integrity across the supply chain, but managing this process is complex. Code-carrying and non-line-of-sight technology are among the technological tools for recording and monitoring the passage of products. Wan et al. [34] conducted a systematic literature review to identify the significant effects of blockchain technology in supply chains such as healthcare and smart city. A lack of transparency and tightly managed information flow impacts a supply chain's effectiveness and data trust among members. The trustworthiness of the data and information provided by business partners within a supply chain or a centralized authority is referred to in this context as data trust.

The efficiency of a supply chain [35] can be increased with the use of accurate data trust in information exchange. Information sharing is recognized as a critical tactic for lowering transaction costs. Supply chain members might share information such as the item's specifications, product conditions, property details, data locations, and even environmental effects [36]. Information exchange is crucial for businesses that do more than make decisions, such as growing profit margins and logistical planning. It is also essential for improving member interaction. The amount of information is proliferating, but it is continually changing from the beginning to the end of the supply chain. To redefine the aspect of trust, blockchain technology can resolve this issue by having just "one trusted ledger". This type of distributed ledger technology can help with reliable information exchange by giving each network member an indelible digital record. As a result, every authorized transaction along the supply chain [37] is documented in a tamper-evidence context. Any deliberate attempt to change the data will be visible and transparent. A supply chain's efficiency and transparency can be improved by combining blockchain technology with IoT and smart devices to transform and automate information collection and dissemination operations digitally. Several researchers have been interested in these potential effects on supply chains [38]. Yet, the overall benefits and limitations of information exchange provided by Blockchain in a supply chain still need to be determined. From the literature review, the major drawbacks of the conventional blockchain-integrated SCM methodologies are

studied, which include low transaction throughput, high time consumption for block generation, lower tracking accuracy, high cost, and complexity. As a result, we will explore and comprehend how blockchain technology can alter how information is transmitted through a supply chain. Table 1 presents the review of SOTA models related to blockchain-enabled SCM in different sectors.

**Table 1.** Review on SOTA models.

| References | State of the Art Models | Purpose and Observations |
|---|---|---|
| [36] | Blockchain in food supply chain | The suggested framework is innovative and is anticipated to help food chain managers determine whether Blockchain is appropriate for their business and/or a larger supplier network. |
| [39] | Vaccine supply chain management | It combines IoT, machine learning and blockchain with SCM, where the security among customers is maintained with the transparency of blockchain. |
| [40] | Blockchain is integrated with SCM for retail sector | From the manufacturer to the final consumer, the substance may be tracked via blockchain technology. As a result, it can guarantee the product's veracity, openness, and trustworthiness throughout the retail supply chain (SC). |
| [19] | Decision aid model for the adoption of blockchain with SCM | It discloses the relationship between cause and effect group issues in the adoption of blockchain and offers structural support to the executives. |
| [41] | Hyperledger blockchain is incorporated with petroleum supply chain | The suggested framework is innovative and is anticipated to help food chain managers determine whether Blockchain is appropriate for their business and/or a larger supplier network. |

## 3. Blockchain Technology

In this technology, the hash value generated by the cryptographic function can be used to identify each block in the data separately. A cryptographic hash function ensures that a single-bit change in the block contents [42] will result in a significant, unanticipated change in the hash code. To build a chain, each new block contains the preceding block's hash code and its actual data. As shown in Figure 5, each block of data comprises the following fields of information:

- The hash value of the previous block,
- Transaction data,
- Nonce message, and
- The hash value of the present block.

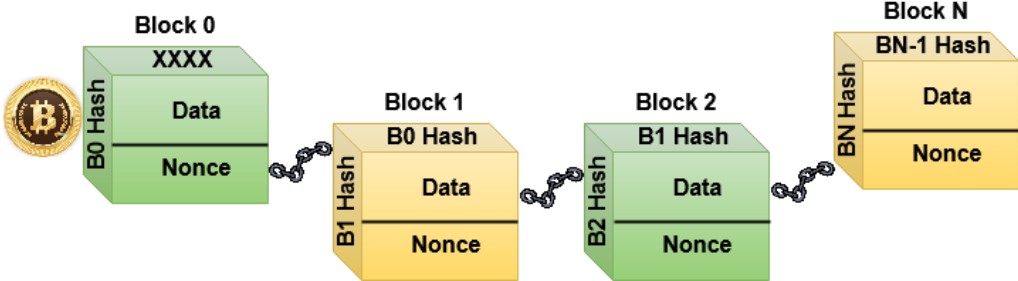

**Figure 5.** Structure of Blockchain.

Blockchain is recognized as a workable way to lessen the complexity of international supply chains and is an appropriate technology for various parts of massive networks. The most complicated transaction is one involving commerce. It involves numerous parties, and the majority of transaction cooperation processes rely on physical documents, which consume a lot of resources, slow down transactions, and obstruct the efficient flow of

commodities. The whole ecosystem will benefit from increased cooperation as a result of the conversion of traditional paper contracts into smart contracts and the realization of the functions of digitalization and independent reading. Distributed ledger technology can be used for effective monitoring, recording, and measurement for preserving environmental sustainability because it will eliminate the potential of illegal activities and content fabrication. The operating system built on the blockchain could enable the digital transformation of manual records and create a secure, open network to facilitate the management of blockchain data, data monitoring, and risk mitigation.

### 3.1. Applications of Blockchain in SCM

A supply chain can be thought of as a planned and methodical network that generates and distributes a certain product to the end user to lower expenses and retain competitiveness in the marketplace [43]. It is made up of many processes, information, and data flows, as well as people, organizations, and additional assets. The supply chain refers to all the steps and parties involved in getting a product from its initial state to the end consumer. This process begins with providing raw materials, their fabrication, transportation through the market, and shipment to the end user. Supply chain operations [44] often involve five parties: vendors, manufacturers, distributors, marketers, and customers. Each party in the supply chain transfers goods to another party following the arrangements made on payments and terms. Blockchains support [45] these contracts, payments, and shipment processes using contemporary supply chain management systems. Optimized supply chain operations are essential for lower costs and a fast production cycle [46]. Moreover, a large number of tasks that span many different domains make up the supply chain management system.

Manufacturers are facing numerous difficulties and are working to find solutions to move forward and expand [47]. It has become essential for manufacturers to meet their goals of obtaining social and economic advantages while upholding practices that avoid adverse effects on the environment. Sustainability development helps manufacturers to adapt to different problems successfully. The information is controlled and stored in a single area using a centralized strategy in the standard manufacturing supply chain. Hence, the entire system is susceptible to error, security breaches, corruption, or assault, because centralization raises the possibility of data loss, which may decrease the chain of supply participants' trust in each other. The centralized manufacturing supply chain also needs a sufficient degree of dependability and transparency in supply chain activities, goods, and procedures. To increase confidence and preserve the current supply chain, it is necessary to strengthen the chain's transparency, information security, trustworthiness, and accountability. By creating a dependable, translucent, accessible, and secured supply chain, blockchain technology [48] promotes sustainable manufacturing practices. This enables how blockchain technology affects sustainability's financial, economic, ecological, and ethical pillars [49]. Typically, finishing the product lifecycle is a fundamental sustainability goal achieved using blockchain technology with reduced obstacles. As shown in Figure 6, data collection, analysis of data, and making decisions are the three basic categories under which technology competencies can be categorized. The application of blockchain technology outside of financial services has primarily been exploratory in nature. Blockchain technology is projected to have some of the most potential non-financial uses in SCM. These industries are good prospects for utilizing blockchain technology and are probably going to pay off early on in the deployment of blockchain. Blockchain technology used in an SCM setting is likely to bring about disruptive changes across all business sectors. As a result, traditional relationship models are already changing, primarily as a result of the transactions' decentralization. Since it is hard to track every occurrence in general, the issue of product delivery operations transparency is important for customers at all levels. Particularly, because the complete network of sellers, marketers, transporters, facilities for storage, and other providers is beyond our control, most end users do not have enough knowledge about the products they purchase and use.

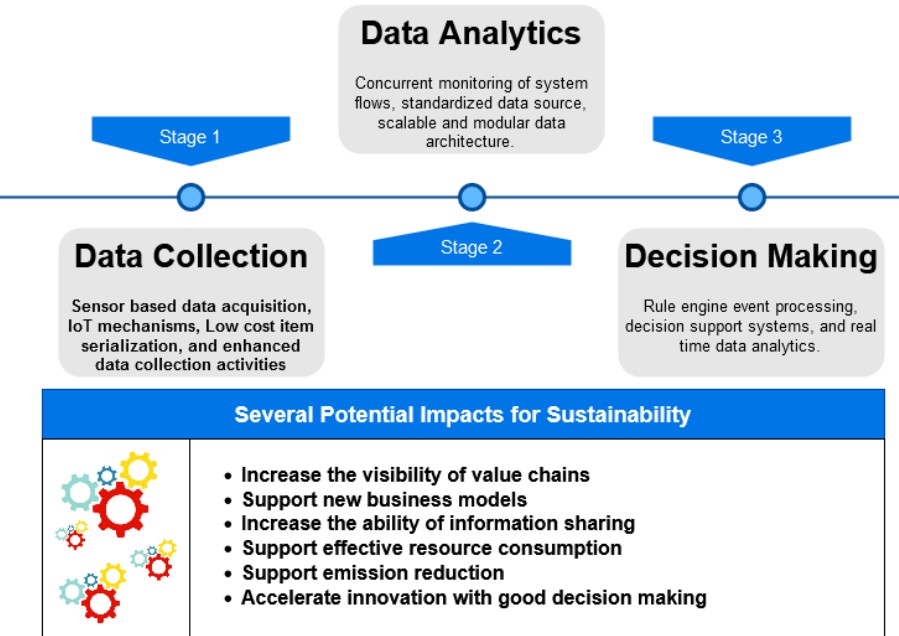

**Figure 6.** Blockchain for sustainability.

### 3.1.1. Fiscal Performance

Manufacturers encounter several difficulties, including high wages, a complex economic environment, and swift changes in consumer demands. Manufacturers should successfully and effectively oversee operations and supply chains to address these issues along with economic sustainability. The fiscal performance [50] assesses the financial efficacy of manufacturers through factors including cash flow, liquidity, and investment potential. Moreover, the expenses related to the goods/services design, material cost, labor cost, machinery cost, and return cost substantially impact manufacturers' financial success. The development of new products should be less expensive, and manufacturers should make better use of resources. Data security and validity are guaranteed by blockchain technology, which lowers the costs incurred by the possibility of tampering.

Additionally, blockchain technology improves information and transaction transparency, fostering stakeholder confidence and openness. Lack of openness and trust increases risk and forces stakeholders to exert more effort to reduce risk, which raises transaction costs. Therefore, blockchain technology ensures data transparency, which improves stakeholder trust, lowers the cost of transactions, and induces profit for businesses [51]. Moreover, the expense of data monitoring and verification is reduced by the ability of blockchain technology to share and validate information in real time across supply chain stakeholders. Manufacturers can improve their financial success by using blockchain technology to analyze competitors' prices and enhance management performance.

### 3.1.2. Quality

The terms "quality" and "production quality" refer to the same thing. The frequency of complaints from customers declines with quality [52]. By making data more accessible and facilitating information sharing among stakeholders, Blockchain may significantly improve quality control and assurance procedures and product design quality. Stakeholders can access information on products, vendors, retailers, suppliers, and supply chain operations because of the capacity to trace information through the entire supply chain [53]. As a result, blockchain technology enables clients to validate the quality of items and guarantees that they adhere to standards, are obtained legally, and are produced following quality control requirements.

### 3.1.3. Ecological Performance and Management

Manufacturers must limit their harmful effects on the environment and optimize their operations and procedures [54]. Sustainable manufacturing techniques are used by businesses to lower their resource usage, boost the efficiency of resources, minimize waste materials, and make goods that can be recycled. The supervision and control of the entire production process—from product development to delivery/disposal—as well as the life cycle of the final product are all part of environmental management. Moreover, it has a substantial positive impact on the economy and society while also protecting the environment. Environmental management practices can benefit greatly from using blockchain technology [55]. Recreating and recalling products is made easier with the data's traceability, exactness, dependability, and real-time access to data. This may lead to resource savings and a drop in emissions. Customers can now determine if products manufactured by manufacturers are sustainable [56] because of the accountability and traceability made possible by blockchain technology, which forces businesses to adopt green practices and minimize emissions.

### 3.1.4. Resource Utilization

Managing resources effectively entails efficiently using resources such as water, electrical power, and materials. Sustainability practices decrease the adverse effects of resources on the ecosystem [57]. These practices include the use of renewable energy in production and the effective use of resources. Blockchain can be used to increase the clean energy sector's sustainability. It also offers distributed strategies for encouraging clean or renewable energy usage. Figure 7 shows the key benefits of using Blockchain for manufacturers' sustainable performance.

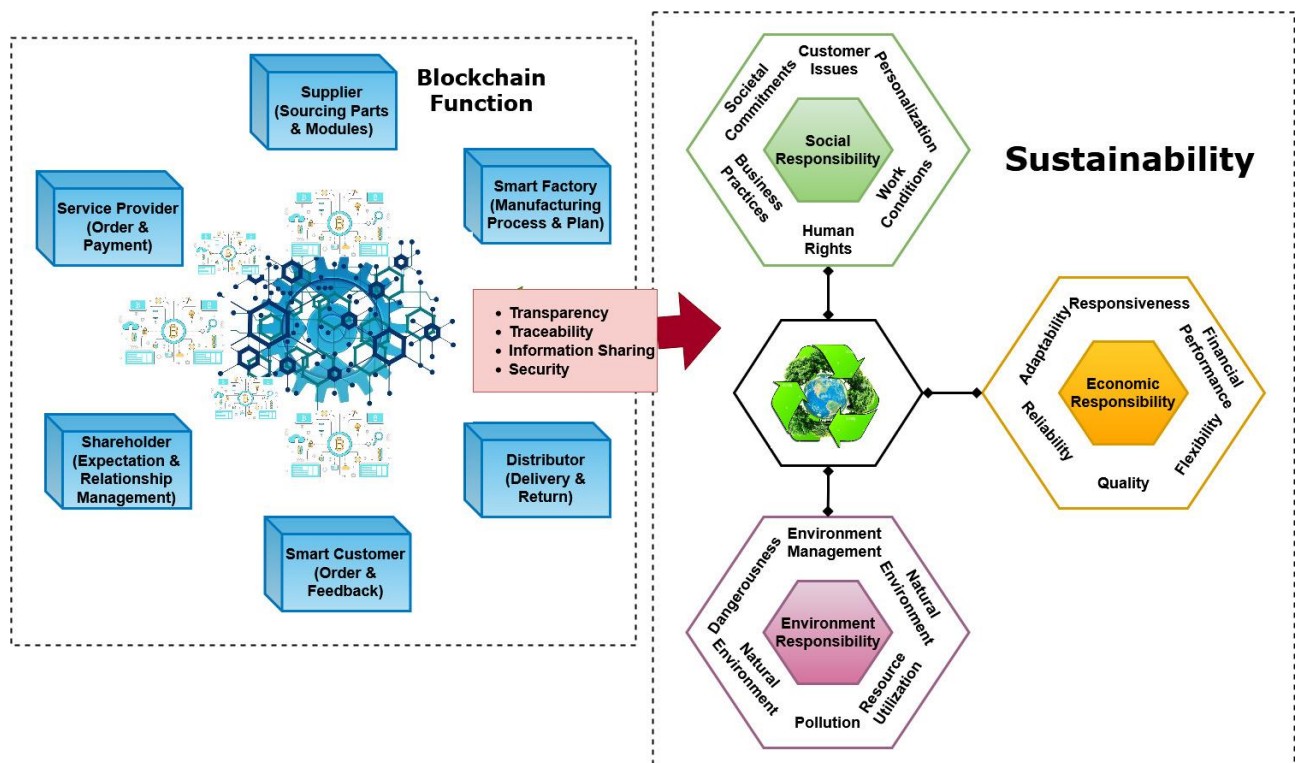

**Figure 7.** Benefits of blockchain for manufacturers' sustainable performance.

### 3.2. Privacy Preservation

Malik et al. [58] implemented a new technique, PrivChain, to safeguard sensitive data using Blockchain. The authors created a smart contract model for automating the process of

provenance verification. Moreover, the Zero Knowledge Proof (ZKP)-based cryptographic model is deployed to verify whether the secret information is true or not. Here, the Pedersen commitment-based secret-sharing scheme is utilized to allow the user to validate the secret message. Chanson et al. [59] utilized a blockchain-based Sensor Data Protection System (SDPS) to ensure the privacy and security of business applications. Tran et al. [60] developed a privacy preservation-based blockchain security model for supply chain management. Peng et al. [61] investigated the specific characteristics of blockchain technology for assuring privacy preservation in IoT systems. Unresolved privacy preservation with high efficiency is a major barrier to the widespread use of blockchain technology. Typically, the Blockchain cannot perform highly computational cryptographic procedures because of its limited computing power. To cut costs, the schemes should also lower the volume of transactions registered in the Blockchain.

As a consequence, effective privacy protection strategies [62,63] are greatly anticipated. Nevertheless, it is difficult to ensure adequate privacy preservation due to the decentralized operation, openness, and inefficiency, which ought to be investigated more thoroughly in the decades to come. The main component of developing diverse blockchain-based applications is the smart contract. Maintaining contract privacy is complicated since it calls for miners to confirm the accuracy of execution results [64,65]. All parties other than the contract creators should not have access to the contract itself or the data created during the execution of the contract. Verifiable computing, or reliable hardware, is used in current methods. They do not fit with Blockchain as evaluated since they are either inefficient or reliant on central authorities.

### 3.3. Challenges of Using Blockchain

Despite the benefits of using blockchain technology, it has some serious challenging issues in adoption and deployment, which are listed in below:

1. Blockchain applications face significant challenges with regard to hardware and energy use. The public key crypto engine and a load dispatcher were coupled to build the blockchain hardware accelerator. The overall transaction speed and output accuracy raise as a result of the load being split among numerous components.
2. Lack of scalability—On that system, many blockchains struggle to accommodate a large number of users. The biggest blockchain platforms saw moderate rates of exchange and increased exchange costs due to a significant increase in clients. Despite the fact that this reality caused extensive research into how to make both these systems and blockchains more equitable, the debate surrounding the suggestions has significantly shifted. The resolution is probably going to take a while.
3. Challenges that still need to be solved include the adaptability of information from blockchain, data security during download, and confidentiality. Peer-to-peer technology have a lot of difficulties executing energy transactions and maintaining information confidentiality. The deployment of blockchain technology was hampered by high resource requirements and transaction costs in energy consumption. A probable answer is the creation of nimble blockchain algorithms enabling instantaneous consensus.

## 4. Supply Chain Management (SCM) Using Blockchain

The fundamental characteristics of openness, verification, automation, and tokenization are only a few of the essential advantages that blockchain technology [66] offers to enhance the coordination and integration of supply chain systems. The main drawback of conventional SCM systems is insufficient end-to-end transparency. Multiple supply chain people [67,68] can exchange real-time information about the location and condition of an object through blockchain technology. With the help of advanced sensors and the prevalence of IoT, it is now possible to track any quantifiable situation, such as the temperature of a product in the cold chain or the availability of technical equipment in a supply chain. Deploying proactive and reactive risk management strategies is also made easier due to enhanced information accuracy.

Moreover, the Blockchain allows people to monitor assets directly from their source by providing a centralized database with readily accessible and unchangeable records. The integrity of assets encompassing both items and technical apparatus is guaranteed by data on provenance to verify legitimacy [69]. This could enforce ethical sourcing and make it possible to identify or stop counterfeit products and other fraudulent processes. Applications may involve tracking down the owner of an object after a sale for purposes of warranty. It also reduces documentation in international trade by guaranteeing the validity of goods documentation, such as those used in customs clearance processes. The supply chain operating as extremely automated based on predefined rules by fusing automation with transparency and validation using smart contracts. Since data and associated actions or choices are spread across the supply chain, this quickens the process and makes coordination easier [70,71]. To make things more explicit, in the event of a machine failure, the machine might contact the supplier to request a spare part, request restoration, and alert downstream stakeholders to potential disruptions. The following Table 2 shows the list of Blockchain's contributions in the SCM field. Table 3 shows the analysis of Blockchain adopted supply chain systems and Table 4 presents the comparative analysis of the blockchain techniques.

**Table 2.** Contributions of Blockchain in SCM.

| Properties | Sub-Domain | Major Contributions |
|---|---|---|
| Consistency [72] | • Customer Assistance<br>• Manufacturers' support<br>• Longevity of shares<br>• Accuracy of predictions | • Increasing information accuracy and reducing errors and mistakes<br>• assisting manufacturing automation<br>• improving the effectiveness of production processes<br>• Increasing the effectiveness of supply chain operations<br>• Increasing the precision of predictions |
| Neutrality [73] | • Design adaptability<br>• Availability of purchases<br>• reliability of the source<br>• flexibility in production<br>• reactivity of the delivery<br>• Selling adaptability<br>• Availability of returns<br>• Adaptability of the supply chain | • Removing mediators as well as directly involving stakeholders<br>• Promoting stakeholder collaboration and communication<br>• Facilitating the sharing of information and resources among stakeholders<br>• Minimizing transactional expenses and time<br>• Improving production techniques<br>• Enhancing operations<br>• Reducing production duration<br>• Reducing client response times<br>• Reducing the duration of maintenance |
| Resilience [74] | • Providers' adaptability<br>• Adaptability in the supply<br>• Flexible production methods<br>• Adaptability in delivery | • Avoiding intermediary firms along with directly involving constituents<br>• Facilitating information and resource sharing among stakeholders<br>• Facilitating interaction as well as collaboration among stakeholders<br>• Promoting the development of innovative processes<br>• Facilitating the decentralization of operations<br>• Supporting quick decision-making<br>• Maximizing collaboration<br>• Increasing the industrial process's flexibility<br>• Improving the ability to make personalized goods<br>• Enhancing responsiveness to transformation and agility |

**Table 2.** *Cont.*

| Properties | Sub-Domain | Major Contributions |
|---|---|---|
| Fiscal performance [75] | • Cost of design<br>• Cost of manufacturing,<br>• Cost of shipment,<br>• Cost of return<br>• Supply chain constraints | • Lowering hazards<br>• Lowering risk-related expenses<br>• Strengthening stakeholder trust<br>• Minimizing transactional expenses<br>• Lowering information tracking and verification expenses |
| Quality [76] | • Quality of goods or services<br>• Performance of suppliers in terms of quality<br>• Quality of production | • Promoting stakeholder information exchange<br>• Facilitating access to information on products, suppliers, retailers, manufacturing, and supply chain operations<br>• Enhancing process surveillance<br>• Simplifying the diagnosis of faults<br>• Allowing buyers to check the product quality<br>• Confirming that products adhere to quality control standards<br>• Ensuring that products are legally sourced and in compliance with the law<br>• Enhancing quality assurance and surveillance |

**Table 3.** Analysis of Blockchain adopted supply chain systems.

| Refs. | Supply Chain Area | Technology Used | Framework Model | Contribution |
|---|---|---|---|---|
| [77,78] | Business application | IoT-integrated blockchain smart contracts | Conceptual framework | It enables effective resource and service sharing. It automated the workflow models of the time-consuming tasks. |
| [79] | SCM | Blockchain-based data storage | Conceptual framework | It helps to reduce the transaction cost and improves the process of executive commitments. |
| [80] | SCM | Blockchain-integrated IoT data storage | Conceptual framework | It assured the properties of data validity and traceability. |
| [81,82] | Manufacturing sectors | Blockchain model | Conceptual framework | It helps to improve the supply chain operations in the manufacturing industries. |
| [83,84] | SCM | Information management | Empirical model | Better adaptability and traceability in the supply chain. |
| [85,86] | Logistics-based business modeling | IoT-integrated Hyperledger technology | Use case model | It helps to improve the efficacy of inbound logistics with better traceability. |
| [87,88] | Supply chain risk management | Artificial Intelligence (AI) based blockchain technology | Conceptual framework | It supports enhancing the resilience of the supply chain system. |
| [89,90] | Supply chain in e-commerce industries | Distributed blockchain technology | Empirical model | It safeguards highly sensitive information by avoiding intermediaries. |
| [91,92] | Financial sector | Blockchain-integrated IoT framework | Conceptual framework | It helps to facilitate market establishment with ensured traceability. |
| [93,94] | Supply chain provenance | Hyperledger technology | Conceptual model | Data validity and accuracy. |
| [95,96] | Business application | Decentralized data storage and security | Conceptual model | It helps to improve the decision-making capability in the supply chain. |

**Table 4.** Comparative analysis of the blockchain techniques.

| Blockchain Techniques | Privacy | Security | Consensus | Speed | Scalability | Transaction Cost | Incentive | Smart Contracts |
|---|---|---|---|---|---|---|---|---|
| Bitcoin [97] | No | No | Computationally Intensive PoW | 7 transactions per sec | No | High | Required cryptocurrency | No |
| Ethereum [13] | Yes | Yes | PoW and PoS | 15 transactions per sec | No | High | Required cryptocurrency | Solidity |
| Hyperledger [98] | Yes | Yes | Multiple approaches | 3000 transactions per sec | Yes | Low | It does not require cryptocurrency | Yes |

The first wave of blockchain technology, often referred to as the Bitcoin blockchain, is made up of cryptocurrency-based network systems. In a Bitcoin network, users are portrayed by the nodes, each with a copy of the same ledger to which blocks of information are sequentially added. The sender, recipient, and transaction amount are only a few examples of the data types that make up a block. Additionally, a block has a hash or message digest that distinguishes it. Each block has a unique hash, calculated from the data in each block, and extra features such as a timestamp. Similar to Bitcoin, Ethereum is a blockchain that runs on cryptocurrencies. It is based on a public network, which can also be used to construct a blockchain with restricted access. Ethereum uses the same PoW protocol as Bitcoin, where the execution of smart contracts allows the decentralized applications to be built on top of Ethereum [99]. It is one of the most crucial features of the Ethereum blockchain. Since it was the very first blockchain platform that launched the idea of smart contracts, it has become quite popular for creating decentralized applications using smart contracts. One of the most cutting-edge blockchain platforms in the Hyperledger series. A blockchain network that is entirely permissioned and tailored for operations requiring private and sensitive information is called Hyperledger Fabric. Granular authorization, secret channels, and zero-knowledge proofs are all supported by hyperledger technology due to their extremely robust privacy and security features. Tables 5 and 6 compares the conventional blockchain techniques based on the different types of parameters, which includes the techniques of Privacy Preserving Blockchain (PPBC) [100], Federated Learning Blockchain (FLBC) [101], Remote Data Integrity Blockchain (RDI-BC) [102], Support Vector Machine (SVM) –Blockchain [103], Access Control Model Blockchain (ACM-BC) [104], clustering BC [105], Forest Fire Detection Blockchain (FFD-BC) [106], Anonymity Access Control Blockchain (AAC-BC) [107], Medical IoT (MIoT)—Blockchain [108], Privacy-Preserving (PMIoT-BC) [109], and Skin Monitoring IoT-BC (SM-IoT-BC) [110].

**Table 5.** Qualitative comparative analysis of the blockchain techniques.

| Techniques | Security | Privacy | Anonymity | Dynamics | Exactness | Energy Utilization | Overhead | Integrity |
|---|---|---|---|---|---|---|---|---|
| PPBC | ✓ | ✓ | X | X | X | X | ✓ | X |
| FLBC | ✓ | X | X | X | X | X | X | X |
| RDI-BC | ✓ | ✓ | X | ✓ | ✓ | X | X | X |
| SVM-BC | ✓ | ✓ | X | X | X | X | X | X |
| ACM-BC | ✓ | ✓ | X | X | X | X | X | ✓ |
| Clustering BC | ✓ | ✓ | X | X | ✓ | X | X | X |
| FFD-BC | ✓ | ✓ | X | X | X | X | X | X |
| AAC-BC | ✓ | ✓ | ✓ | X | X | X | X | X |

**Table 5.** *Cont.*

| Techniques | Security | Privacy | Anonymity | Dynamics | Exactness | Energy Utilization | Overhead | Integrity |
|---|---|---|---|---|---|---|---|---|
| IoT-BC | X | ✓ | X | X | X | X | X | X |
| MIoT-BC | ✓ | ✓ | X | X | X | X | X | X |
| PMIoT-BC | ✓ | ✓ | X | X | ✓ | ✓ | X | X |
| SM-IoT-BC | ✓ | ✓ | X | X | X | ✓ | X | X |

**Table 6.** Comparative analysis of the blockchain techniques using different parameters.

| Techniques | Con | Avail | Eff | Acc | Cost | Scal | Run Time | Rel |
|---|---|---|---|---|---|---|---|---|
| PPBC | X | X | X | X | ✓ | X | X | X |
| FLBC | X | X | ✓ | ✓ | X | ✓ | ✓ | X |
| RDI-BC | X | X | X | X | X | X | X | X |
| SVM-BC | X | X | ✓ | ✓ | X | X | X | X |
| ACM-BC | X | ✓ | ✓ | ✓ | X | X | X | X |
| Clustering BC | X | ✓ | ✓ | ✓ | X | X | X | X |
| FFD-BC | ✓ | X | X | ✓ | X | X | X | X |
| AAC-BC | X | X | X | X | X | X | X | X |
| IoT-BC | X | X | ✓ | X | X | X | X | ✓ |
| MIoT-BC | X | ✓ | ✓ | X | X | X | X | X |
| PMIoT-BC | X | X | ✓ | ✓ | X | X | X | X |
| SM-IoT-BC | ✓ | X | X | X | X | ✓ | X | X |

Con—Confidentiality, Avail—Availability, Eff—Efficiency, Acc—Accuracy, Cost, Scal—Scalability, and Rel—Reliability.

## 5. Results and Discussion

A comparative analysis has been carried out to validate the efficacy of both conventional and blockchain-enabled SCM strategies. For this assessment, the parameters such as latency, execution time, efficiency, and security level have been considered. As shown in Figure 8, the latency is compared with the Ethereum [111] and hyper ledger smart contract [112] technologies used in SCM systems. Typically, latency is defined as the delay of processing, which must be reduced to achieve more incredible system performance. The results indicate that the hyperledger smart contract provides reduced latency compared to the Ethereum models.

Consequently, the efficiency of both conventional and blockchain-enabled SCM frameworks are compared, as shown in Figure 9. The overall system efficiency can be greatly increased when the SCM is incorporated with blockchain technology due to its resilience, consistency, integrity, and quality properties. Moreover, the execution time of the traditional client/server and blockchain-enabled frameworks are compared concerning the number of records, as shown in Figure 10. Then, the security level of SCM is also validated and compared, as shown in Figure 11. The overall analysis indicates that the supply chain process of any industrial application can be improved when it adopts blockchain technology.

Figure 12 compares the blockchain index size of various techniques used in the medical healthcare supply chain systems, where the index size is estimated concerning varying numbers of records. Consequently, Figure 13 compares the efficiency of the blockchain techniques used in the healthcare chain systems. These parameters are computed and

compared in this study to determine the efficiency and performance of various blockchain techniques used in supply chain systems. Overall, the study results indicate that the blockchain index size has decreased with an increased number of processing records. Moreover, it is essential to reduce the blockchain techniques' time and latency to ensure better system performance outcomes.

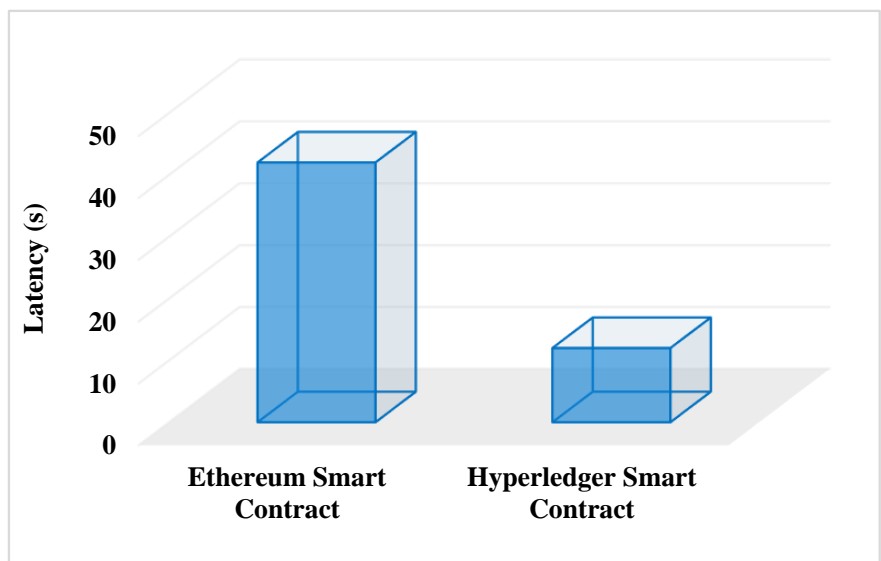

**Figure 8.** Latency.

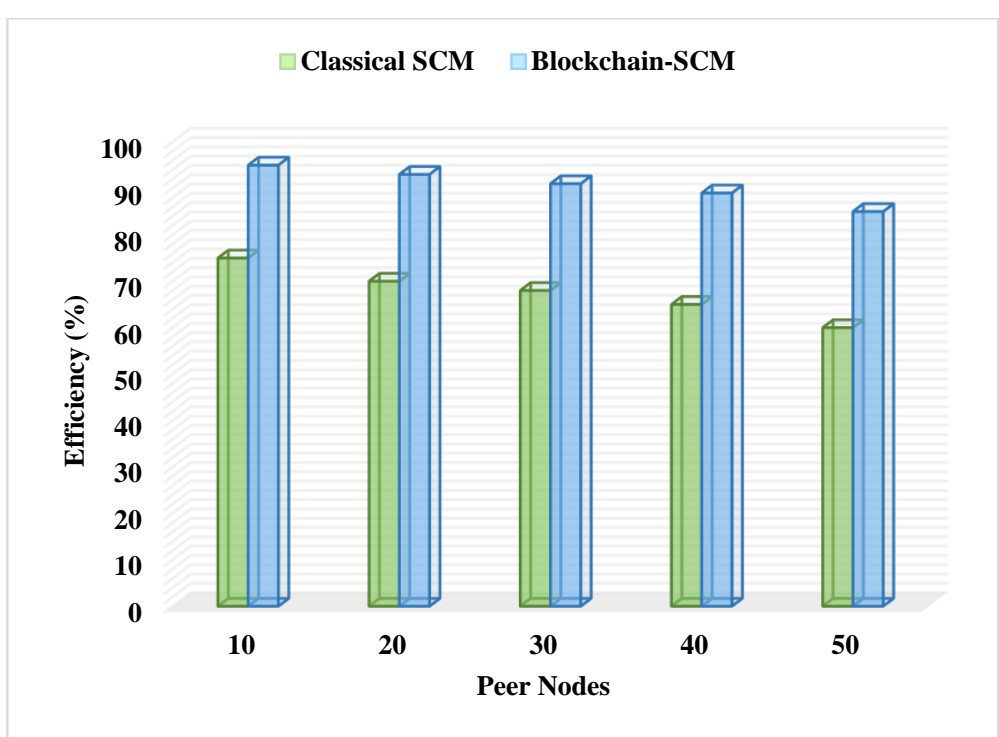

**Figure 9.** Efficiency.

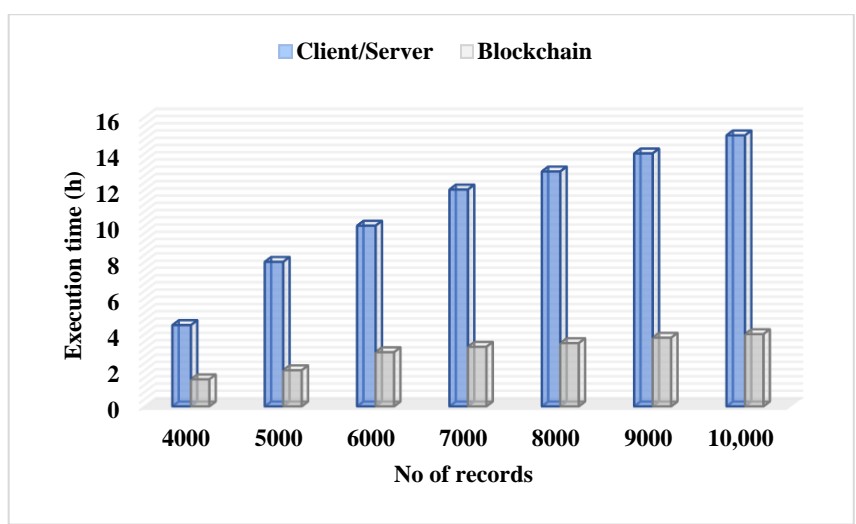

**Figure 10.** Execution time.

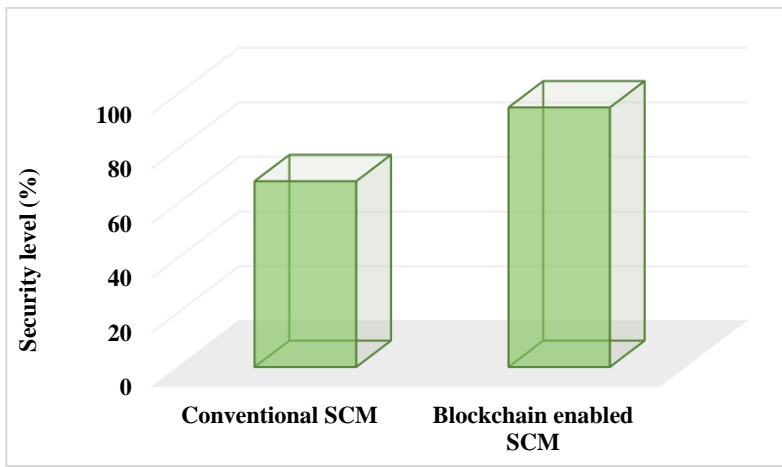

**Figure 11.** Security level.

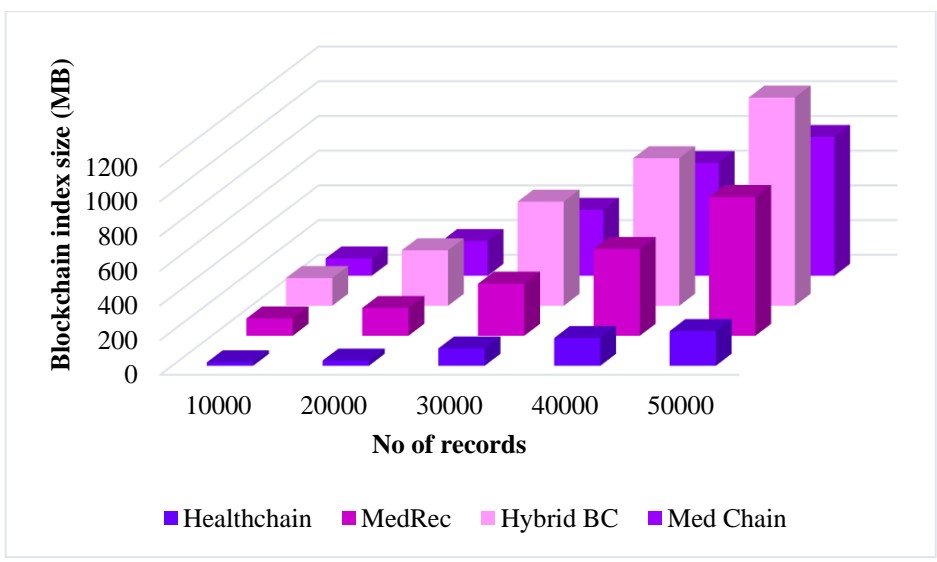

**Figure 12.** Analysis based on blockchain index size.

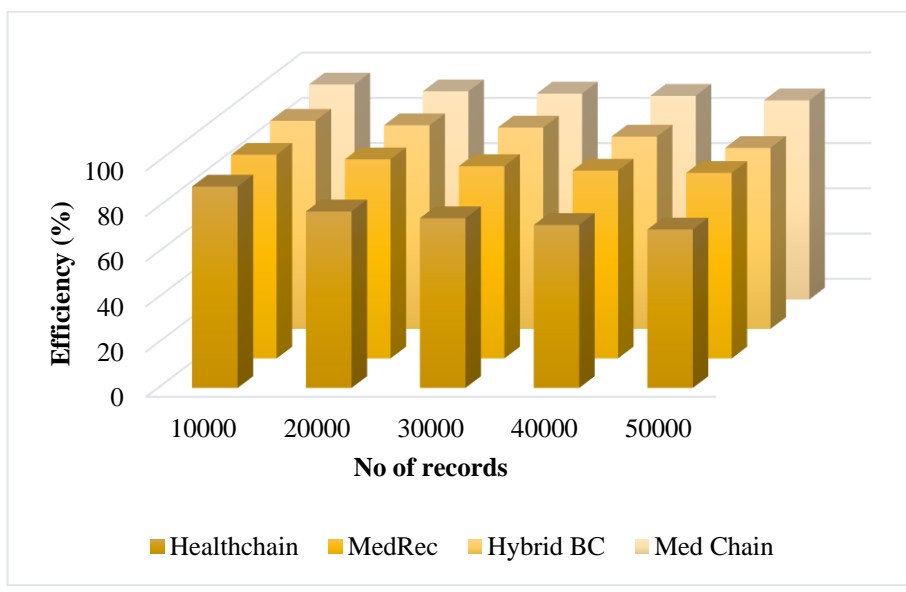

**Figure 13.** Efficiency analysis.

## 6. Challenges and Recommendations

Blockchain systems should be versatile to enable the design of verification algorithms to incorporate various supply chain applications. To update the blockchain ledger, participants must apply the verification algorithms to approve or deny the received blocks and transactions. Typically, supply chain systems require product-specific attributes or features and verification codes for validating every transaction. Hence, it is essential to determine the verification models and transaction history for different products before the validators add a legitimate record to the blockchain ledger. Moreover, the significant challenges in adopting blockchain techniques for SCM are needed for more transparency, sustainability, provenance, and privacy. In addition, some other factors, such as complex data collection and management, unsuccessful system implementation, and limited knowledge about emerging trends, are also considered critical challenges of blockchain adoption. We found a significant number of publications in the studied literature, some of which are experimental and others that left blockchain technology beyond the scope. According to the report, there is a growing interest in adopting blockchain technology to address numerous supply chain concerns. Blockchain applications in real-world supply chains have yet to be reported, but they may help us determine this technology's true potential and find solutions to current supply chain issues.

## 7. Conclusions

The purpose of this article is to undertake a thorough analysis of the value of employing supply chain systems that can use blockchain technology. Blockchain technology has recently being widely used by businesses to manage their supply networks. Therefore, the main objective of contemporary blockchain projects is to increase supply chain transparency. For instance, the less complicated requirements for product complexity in medical supply chains can already be fully reproduced on the Blockchain. No system is capable of accurately mapping production processes, auditing all assets, or implementing dynamic supply chain changes that aim to increase transparency. Even though the characteristics of raw materials, intermediate components, finished goods, and transformational activities are all very different from one another, they all inevitably interact or come together at some point in complex industrial supply chains. A strong smart contract-based architecture could be a significant step towards bringing the advantages of blockchain technology to complex industrial networks. In order to examine the technical implementation aspects of SCM systems made possible by Blockchain, this article use rigorous techniques. For the purpose

of ensuring the security of industrial applications, we largely concentrated on developing SCM systems with blockchain capabilities. In this study, the special characteristics of blockchain technology have been used to analyze its major results in the SCM. Several security criteria are utilized to compare and validate the efficacy of the blockchain approaches used in SCM. Moreover, in order to compare the traditional SCM and blockchain-enabled SCM systems, common factors such as latency, execution time, efficiency, and security level are taken into account. Blockchain is a new revolutionary technology with unmatched benefits including decentralized governance, transparency, autonomy, and security. The literature on extending blockchain technology to the business applications is still sparse, and studies on the areas of concern in industrial sector are extremely scarce, despite the fact that many recent studies have examined the practical uses and challenges of blockchain technology in the supply chain field. We made a courageous attempt to research the use of blockchain technology in SCM. This review's conclusion is that the Blockchain is thought to be the most effective solution for enhancing the SCM systems used in all industrial applications, including healthcare, automotive, and other fields. By using a new cryptographic integrated blockchain technique for healthcare applications, this work can be improved in the future. In order to establish a blockchain-based network system and address business points of discomfort in the industrial supply chain, our research will investigate how to best utilize blockchain technology.

**Author Contributions:** Conceptualization, M.U. and S.S.; methodology, M.O.; software, S.U.A. and M.A.; validation, M.U., S.S. and A.O.K. (Adil O. Khadidos); formal analysis, M.O. and M.A.; investigation, A.O.K. (Alaa O. Khadidos) and A.O.K. (Adil O. Khadidos); resources, S.U.A.; data curation, S.S.; writing—original draft preparation, M.U. and S.S.; writing—review and editing, S.S and M.O.; visualization, S.U.A. and M.A.; supervision, M.U.; project administration, S.S. All authors have read and agreed to the published version of the manuscript.

**Funding:** This research was supported by Princess Nourah bint Abdulrahman University Researchers Supporting Project Number (PNURSP2023R97), Princess Nourah bint Abdulrahman University, Riyadh, Saudi Arabia.

**Data Availability Statement:** Data will be made available upon request.

**Acknowledgments:** Princess Nourah bint Abdulrahman University Researchers Supporting Project Number (PNURSP2023R97), Princess Nourah bint Abdulrahman University, Riyadh, Saudi Arabia.

**Conflicts of Interest:** The authors declare no conflict of interest.

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
