# Peer review of "From Hype to Reality: Unveiling the Promises, Challenges and Opportunities of Blockchain in Supply Chain Systems"

_sustainability, doi:10.3390/su151612193_

Round 1
Reviewer 1 Report
The use of blockchain in supply chain management is a remarkable study in terms of both academic and sectoral applications. Although I find the study valuable in general terms, I think that the following arrangements should be made:
-In academic studies, the hierarchy of "motivation, limitation and contribution" is important in the Introduction. I perceived that not much attention was paid to this in this study. While the importance (motivation) of the use of blockchain in supply management systems is well defined, the deficiencies (limitatiob-n) identified in the studies on these issues are not mentioned. In addition, I am thinking of rewriting the additives of the study by expanding and elaborating.
-The technical information about Blockchain and SCM given in the introduction can be taken to the "Related Works" section. That section can be rearranged like "Knowledge&Related Works" or in another way.
-It is necessary to give detailed information about the Blockchain infrastructure used in the study (physical equipment, virtual machines, etc.) and how they are used.
-The tables given in Chapter 4 are good for a better understanding of the study. But can the page structure be adjusted horizontally to make the article more visual and clearer for the reader?
Author Response
We are thankful to all the reviewers for your precious comments which have helped us improve this manuscript's quality. It has further enriched our understanding and helped organize our presentation better in this version of the manuscript. Point-by-point responses to all the comments have been given in the attached file. Following the comments from the Reviewers, respective changes have been included in the revised manuscript and all changes have been highlighted with comments.

Reviewer 2 Report
Reviewers’ Comments to the Author(s):
Reviewer 1:
1. Mention the impacts of integrating blockchain in sustainability management in abstract. 2. Add SOTA models in the related works with appropriate citations. 3. Section 3 should be elaborate a little bit more with the operations of blockchain. 4. Discuss about other applications, advantages, impacts and etc of using blockchain for sustainability management. 5. Describe the properties of blockchain in SCM with citations. Why there is no reference for Table 1 and Table 3? 6. Update the conclusion with overall study analysis and investigation. Also, state the future work clearly based on the findings. 7. What is the inference from the study? 8. Check the reference list some of them are incomplete.Author Response
We are thankful to all the reviewers for your precious comments which have helped us improve this manuscript's quality. It has further enriched our understanding and helped organize our presentation better in this version of the manuscript. Point-by-point responses to all the comments have been given in the attached file. Following the comments from the Reviewers, respective changes have been included in the revised manuscript and all changes have been highlighted with comments.

Reviewer 3 Report
The paper is relevant and topical. It captures the essence of most of the literature on the intersection of blockchain and SCM. The nature of the paper is a review paper with an overview of blockchain applications in SCM provided, different blockchain-enabled supply chains, and techniques as well. Some comparative analysis has also been done. Use-cases and case studies can be added in the future enhancements of the work.
Good language, presentation and flow devoid of major grammatical errors
Author Response

(The authors gave the same response as above.)

Reviewer 4 Report
Please read the attachment

Author Response

(The authors gave the same response as above.)

Round 2
Reviewer 1 Report
With the amendments suggested in the previous version of the article, your work has become able to contribute more to the literature. In particular, the motivation and contribution parts of the study were better understood. Graphics and pictures have good image quality and are used in the article for fluency. It was appropriate to refer to more recent articles in the references section. I wish you success in your future work.